# The Evolution of Complex Multicellularity in Land Plants

**DOI:** 10.3390/genes15111472

**Published:** 2024-11-14

**Authors:** Hossein Madhani, Arsham Nejad Kourki

**Affiliations:** 1School of Life Sciences, University of Nevada Las Vegas, Las Vegas, NV 89154, USA; 2The Francis Crick Institute, London NW1 1AT, UK; 3Department of History and Philosophy of Science, University of Cambridge, Cambridge CB2 3RH, UK

**Keywords:** evolutionary transitions in individuality, major evolutionary transitions, multicellularity, land plant evolution, sexual life cycle, evolution of complexity

## Abstract

The evolution of complex multicellularity in land plants represents a pivotal event in the history of life on Earth, characterized by significant increases in biological complexity. This transition, classified as a Major Evolutionary Transition (MET), is best understood through the framework of Evolutionary Transitions in Individuality (ETIs), which focuses on formerly independent entities forming higher-level units that lose their reproductive autonomy. While much of the ETI literature has concentrated on the early stages of multicellularity, such as the formation and maintenance stages, this paper seeks to address the less explored transformation stage. To do so, we apply an approach that we call Transitions in Structural Complexity (TSCs), which focuses on the emergence of new units of organization via the three key evolutionary processes of modularization, subfunctionalization, and integration to the evolution of land plants. To lay the groundwork, we first explore the relationships between sex, individuality, and units of selection to highlight a sexual life cycle-based perspective on ETIs by examining the early stages of the transition to multicellularity (formation) in the sexual life cycle of the unicellular common ancestor of land plants, emphasizing the differences between the transition to multicellularity in eumetazoans and land plants. We then directly apply the TSC approach in this group, identifying key evolutionary events such as the distinct evolutionary innovations like shoot, root, vascular systems, and specialized reproductive structures, arguing that bringing these under the broader rubric of TSCs affords a degree of explanatory unification. By examining these evolutionary processes, this paper provides a new perspective on the evolution of multicellularity in land plants, highlighting both parallels and distinctions with the animal kingdom.

## 1. Introduction

The history of life on Earth is marked by a series of pivotal evolutionary events, often categorized as Major Evolutionary Transitions, henceforth METs [1]. METs were originally defined as events characterized by two processes: (1) formerly independent entities coming together to form higher-level units of organization, resulting in them only being able to reproduce as parts of this greater whole—in other words, losing their reproductive autonomy and thereby ceasing to be units of selection; and (2) the evolution of new ways in which information is processed and transmitted over generations. Within this general framework, a leading approach has been to focus only on the former process in the interest of theoretical coherence. This is the so-called Evolutionary Transitions in Individuality approach—ETI for short. The ETI approach trims the list of METs to only those where a new unit of selection has evolved from formerly independent lower level units and excludes events that would otherwise count as METs based on the second criterion of the original MET formulation, such as the evolution of sex or the evolution of language. Uncontroversial ETIs, therefore, include the evolution of the first protocell from pre-existing replicators or metabolic cycles, the evolution of the first eukaryotic cell from pre-existing prokaryotes, the evolution of multicellularity, and the evolution of eusociality.

Of these four ETIs, the evolution of multicellularity is plausibly the best-studied so far. This was an evolutionary step that revolutionized biological complexity, leading to the emergence of complex multicellular organisms [2,3,4,5,6,7,8,9]. Within the ETI approach, several authors have distinguished between earlier and later stages of a transition. Notably, Bourke [8] distinguishes between three stages: *formation*, *maintenance*, and *transformation*. In the evolution of multicellularity, the formation stage generally includes the initial steps taken by unicellular organisms towards (they can, of course, never go beyond these first steps and retain the simplest forms of multicellularity. It is no coincidence that these simplest forms are so frequent across the tree of life: the formation stage is readily achieved, which has even been shown experimentally in several phyla [10,11,12]) forming a loosely-organized multicellular collective, such as failure to dissociate after cell division or even after nuclear division, or the aggregation of independent unicellular organisms into a multicellular form, such as what we see in *Dictyostelium* [7]. The maintenance stage is what makes the difference between the simplest multicellular forms (which are often transient, e.g., *Dictyostelium*) and more complex ones. It has long been recognized that a collective of conspecifics, even if closely related, faces a persistent risk of what has been called ‘subversion from within’—that is, the emergence of conflict between free-riders and cooperators in the collective and the eventual collapse of the cooperative regime [13,14]. Accordingly, the maintenance stage is characterized by the evolution of mechanisms of conflict mediation, ranging from reproductive bottlenecks to active policing of defectors and the sequestration of germlines at an early developmental stage [2].

The ETI literature as a whole is less clear on what the transformation stage is, but the general idea behind it is that it consists of what follows the maintenance stage and results in the transformation of the stable social group into a full-fledged, complex multicellular organism. The focus of the ETI literature on the first two stages has resulted in an explanatory gap around the transformation stage: (though some authors forego sharply distinguishing between the maintenance and transformation stages altogether, which makes this explanatory gap disappear from sight; e.g., [15]) what exactly are the processes that result in the emergence of a complex multicellular organism? In a previous publication (as well as a work in progress), Nejad Kourki [9] has attempted to fill this explanatory gap in the context of early metazoan evolution by drawing on hierarchy theory [16,17,18] and proposing a structuralist approach to ETIs. Here, we seek to extend this framework to the domain of land plants. The overarching goal is to provide a comprehensive understanding of the evolution of multicellularity in land plants using the theoretical insights previously detailed for animals. This transition in animals has been at the forefront of research, culminating in comprehensive theoretical frameworks that elucidate the processes underpinning multicellularity. Land plants, with their unique evolutionary trajectories, present both challenges and opportunities for the application of existing theoretical paradigms. While the foundational steps towards multicellularity might share similarities between plants and animals, the subsequent evolutionary nuances, driven by distinct ecological pressures and evolutionary histories, differ substantially [19,20].

## 2. Transitions in Structural Complexity

The theoretical approach developed by Nejad Kourki revolves around three abstract processes that underpin changes in the structural complexity of multicellular organisms over evolutionary timescales. While they were initially tailored to metazoan evolution, we aim to show here that they are equally applicable to the evolution of complex multicellularity in the plant kingdom. This approach, which we term **Transitions in Structural Complexity** (while the paper cited here presents its theoretical approach as an extension of the ETI approach, Nejad Kourki is currently developing a more comprehensive elaboration of this theoretical approach under a new label—namely, Transitions in Structural Complexity. We will use this label in this paper), focuses on the emergence of new units of organization (which includes units of selection) and rests on three principal processes: *modularization*, *subfunctionalization*, and *integration* [9]. These processes underpin the multiplication, differentiation, and emergence of structures and functions within complex multicellular organisms (Figure 1). For instance, while the initial stages of multicellularity might be characterized by basic coherence and division of labor among cells, the subsequent evolution towards complex multicellularity involves processes like the emergence of modular organization of structures (modularization), the differentiation of specific structural units (subfunctionalization), and the integration of different structural units into new ones. These processes collectively transform the organism from a simple colony of cells to a highly integrated, adaptive, and complex entity, whose basic biological functions are carried out via integrated units at the multicellular level rather than in a collective manner at the cellular level. Thus, an important part of the explanatory work of TSC comes down to the transfer of organismic functions from lower to higher levels of organization (Table 1).

For land plants, this evolutionary narrative, while sharing foundational elements, diverges in unique ways. The challenges faced by early land plants, from water and resource acquisition in terrestrial environments to gamete dispersal without an aqueous medium, necessitated innovative solutions. The resulting evolutionary landmarks, such as the development of vascular systems, specialized reproductive structures, the alternation of generations, and the evolution of seed and wood, highlight the distinct trajectory of plant multicellularity. By appreciating the interplay between genetic factors, structural features, and developmental processes, we can better contextualize the evolution of multicellularity in land plants, drawing parallels and distinctions from the animal kingdom. This paper will delve into these nuances, offering an exploration of multicellularity from a land plant perspective.

The three core processes underlying the evolution of structural complexity—modularization, subfunctionalization, and integration—collectively result in higher vertical complexity, marked by the advent of new intermediate organizational levels and the consolidation of the higher level. They also lead to increased horizontal complexity at intermediate levels, like organs and organ systems in multicellularity, and conversely, decreased horizontal complexity at the lower level in a parallel evolutionary process variously dubbed de-Darwinization [21] or machinification [22]—e.g., cell types in more complex multicellular organisms losing many of their parts in favor of focusing on specific functions. It should be noted that specialization and differentiation, often used interchangeably, have distinct connotations here. Differentiation specifically pertains to cells becoming more specialized, while specialization is a broader term encompassing differentiation at the cell level and subfunctionalization at intermediate levels. Building on this theoretical framework, the subsequent sections will elucidate how these processes underlie the evolution of multicellularity in land plants, drawing parallels with the animal kingdom and highlighting the unique trajectories of plant evolution.

## 3. TSCs, Sex, and Multicellularity

### 3.1. Complex Multicellularity

When broadly defined as the ability to sustain a community of cells, the transition to multicellularity is estimated to have evolved up to 45 times in the Tree of Life [23,24,25]. However, complex multicellularity, defined as having several levels of organization and a considerable degree of complexity within each level and characterized by cell-to-cell signaling and a heritable phenotype [26], depending on how strict or loose our criteria are, is evident in at most 11 transitions, and uncontroversially in the 2 most familiar ones—the evolution of animals and plants; thus, the evolution of multicellularity in land plants represents one of these transitions across eukaryotes. With more permissive criteria, we may also include once in Amoebozoa (dictyostelids), three times within fungi (including chytrids, ascomycetes, and basidiomycetes), and six occurrences across the three major photosynthetic eukaryotic clades—two occurrences each in rhodophytes, stramenopiles, and chlorobionta, with the land plants representing one of the two chlorobionta instances [26,27,28].

### 3.2. Sex and Multicellularity

The emergence of complex multicellularity in land plants, as in other life forms, originated within a sexual life cycle [8]. This suggests that the evolution of complex multicellularity is fundamentally linked to the sexual life cycle, with the transition to sexual reproduction marking the initial step towards complex multicellularity in the evolutionary history of these 11 clades. In this vein, the evolution of multicellularity can be viewed as an augmentation of the sexual life cycle, enhancing the fitness of individuals within this cycle. To delve into this perspective, we should examine the origins of multicellularity in the two most paradigmatic cases—land plants and eumetazoans. This requires defining the type of sexual life cycle in the unicellular common ancestors of these significant clades. Traditionally, there are three types of sexual life cycles; haplontic, diplontic, and haplodiplontic, categorized based on the nuclear phase of the predominant cell(s) in the cycle (See Figure 2). In unicellular organisms, the cell is normally referred to as the “individual,” the primary level at which selection operates from the ETI perspective. Identifying the sexual life cycle type of the common ancestor of a multicellular organism helps us pinpoint where selection is acting, leading to a transition in individuality. For instance, in Animalia, ETI theory posits that the incomplete dissociation of multiple units (individuals) produced by the zygote’s mitosis in the unicellular diplontic life cycle (Figure 2c) of the common ancestor led to the formation of social groups, marking the first step of ETI.

Another important aspect of this perspective is defining the boundaries between individuals in the cycle, which itself is very difficult and complicated. The two concepts of sex and individuality are in constant tension, as per the definition of individuality, which is equated with functioning as a unit of selection, while sex mixes genes from different individuals (unit of selection) at the boundary between generations [13]. Notably, in a diplontic unicellular eukaryote, the ancestral state of multicellularity in animals, individuality begins with the dissociative mitosis of the zygotes, separating the fate of these sister cells (Figure 2c). However, in diplontic multicellular organisms like eumetazoans, individuality commences with fertilization. This distinction is pivotal for understanding a critical, yet often overlooked, aspect in the literature on social evolution: during ETI, the cells that coalesce, or rather, do not dissociate post-production, already share the same sexual life cycle as they all result from the same fertilization event, forming a family of cells rather than a society. This concept also applies to eusocial insects, where a family of insects produces a colony. In multicellular contexts, however, each multicellular individual may have its own life cycle, sharing it only during the gamete phase for the purpose of reproduction, which involves the exchange of gametes between typically two separate sexual life cycles. This pattern appears to occur across all levels of the sexual life cycle, where different individuals sharing a sexual life cycle tend to cooperate, forming higher levels of organization. Through this cooperation and interdependence with other units within the sexual life cycle, the fitness of all lower-level units involved in the cycle is enhanced. Recognizing this is crucial for addressing fundamental questions such as the reason behind the tendency of life in forming higher levels of organizations among sexual reproducers (11 independent times). How is the fitness of units occupying the higher level of organization coupled with that of the lower units? How can this reconcile the tension between sexual reproduction and ETIs? And why is a unicellular phase, or bottleneck, an indispensable part of all complex multicellular organisms?

A universal feature of multicellular organisms is the presence of a unicellular bottleneck, which is hypothesized to have evolved because it reduces genetic variance within the multicellular organism and increases variance among individuals [29,30]. Another hypothesis, which seems more reasonable, is that the unicellular bottleneck in multicellular organisms is a consequence of their secondary origin from sexual unicellular ancestors [29], a relic of an ancestral life cycle co-opted into the developmental program of the multicellular organism [31]. A more descriptive answer to the universal presence of a unicellular bottleneck is because multicellular organisms are sexual reproducers like their unicellular ancestors, and sexual reproduction needs a unicellular phase for the fusion of genetic materials from, generally, two different individuals. In other words, the multicellularity evolved as an extended complexity within a sexual life cycle of multiple unicellular individuals that already have shared the same sexual life cycle at the zygotic stage; so, for the higher organization to maintain the ability to reproduce after transition, it has to retain unicellular phases, such as gametes, zygotes, and spores.

Let us explore the evolution of multicellularity in land plants in this regard. It is not clear whether the last common ancestor of land plants and streptophyte algae was a unicellular or multicellular organism [32,33,34]. An answer to this question is crucial for identifying where the first Evolutionary Transition in Individuality (ETI) began to occur, and phyletic distribution of the haplobiontic–haploid life cycle, in which only the haploid phase is multicellular, across Chlorobionta, also known as Viridiplantae or green plants, shows that transition to multicellularity occurred before the independent evolution of diplobiontic life cycle in embryophytes and *Ulvophyceae* [32]. Recent phylotranscriptomic analysis show that transition to multicellularity in streptophytes first happened around 1 billion years ago in the common ancestor of Phragmoplastophyta (including embryophytes) and *Klebsormidiophyceae algae* [35]. The earliest credible records of Archaeplastida, specifically *Bangiomorpha pubescens* and *Proterocladus antiquus*, are traced back to fossils from the early Neoproterozoic era (1078–940 Ma) in Liaoning, North China [36]. These records show branching multicellular thalli with a differentiated holdfast, bearing a strong resemblance to siphonocladalean green algae [36]. Yet, the simplicity of this structure complicates the exclusion of convergent evolution. Older fossils are predominantly aligned with cyanobacteria and are often indistinguishable from filamentous cyanobacteria [37]. Accepting the mainstream narrative of three independent ETIs in plant evolution, we can infer that the multicellularity observed in contemporary land plants has its roots in the common ancestor of green plants. The simplest form of multicellularity in green plants is observed in the loosely connected sticky colonies of *Prasinodermophyta*, a recently discovered phylum within *Viridiplantae* [38]. The ancestral unicellular sexual life cycle of the green plant common ancestor was likely haplontic, where mature individuals of the cycle are haploid cells, similar to *Volvox* and *Chlamydomonas*.

Transition to multicellularity in green plants began with the formation of the gametophyte in their common ancestor, leading to a haplontic multicellular life cycle. This cycle eventually evolved to include the sporophyte, which significantly enhances fitness by increasing the number of spores produced per generation. In the multicellular haplontic life cycle, zygotes produced via sexual reproduction undergo meiosis following fertilization, typically resulting in four spores. However, imagine a scenario where a simple change causes the zygote to undergo mitosis, producing two sister cells, each retaining the ability to undergo meiosis. Consequently, this process would yield eight spores instead of four, directly boosting the fitness of a multicellular individual with a haplontic life cycle. Any further increase in the number of mitotic divisions before meiosis would, in principle, exponentially enhance fitness. This increase in the number of produced progeny per fertilization is likely an adaptation to terrestrial environments where available water is limited for water-dependent fertilization [39]. Therefore, the evolution of the sporophyte and the alternation of generations in land plants can be seen as a vertical increase in the complexity of the diploid phase of the life cycle of a haploid multicellular ancestor that has already undergone a transition in the haploid phase of its life cycle. So, evolution of the sporophyte is not a transition in individuality per se; however, it adds a new level to the complexity of the life cycle of a haploid multicellular organism and an independent unit of selection (Figure 3).

## 4. The Evolutionary Processes of TSC

### 4.1. Modularization in Land Plants

Within the TSC approach, modularization can be easily seen as the most fundamental evolutionary process: it is the process that generates multiple units of organization within the emerging or existing higher-level unit, either via the multiplication of existing lower-level units or the subdivision of a unit into smaller, constituent units. The first instance of modularization in the evolution of any multicellular lineage is the emergence of the multicellular collective itself, which typically happens through the multiplication of unicellular organisms that do not subsequently dissociate. But it is the same general process that gives rise to units of organization sandwiched between the level of the individual cell and the multicellular organism as a whole. Examples from the animal kingdom include the emergence of segments and their associated structures ranging from nephridia and limbs to eyes and ribs, as well as integumentary appendages such as scales, feathers, hair, chaetae, etc. Modularization is fundamental because it provides the raw material for the other two processes in generating structural complexity. We now turn to paradigmatic cases of modularization in land plants.

Liverworts present clear examples of modularity in their gametophytes. The leafy liverworts of the *Jungermanniales* order, the *Noteroclada* species from *Pelliales*, and members of the *Porellales* order all exemplify this modular architecture. *Marchantia*, a genus familiar to many botanists, offers another example. Within its intricate anatomy of thalloid Liverworts like *Marchantia*, modularity is evident in antheridiophore and archegoniophore. These are the male and female reproductive organs, respectively, which house the antheridium and archegonium—modular sexual organs that produce male and female gametes. *Marchantia*’s modular architecture does not end here: gemmae cups, specialized for asexual reproduction, dot its body. Each cup, cradling gemmae that can form new plants, epitomizes a modular strategy for rapid propagation. Then, there are the rhizoids, ubiquitous among the gametophytes of bryophytes and non-seed vascular plants, that also develop in some of the streptophyte algae, such as *Chara* and *Spirogyra* [41]. Though not homologous to the true roots (though possibly homologous to root hairs and co-opted from one generation to the other) in the sporophytes of the vascular plants, rhizoids also follow the modular theme. Their primary role is anchorage, binding the plant to its substrate. Nevertheless, they also absorb water and nutrients. This modular design equips them to tap into resources efficiently, especially useful in habitats where nutrients are hard to come by.

Modularity is also apparent in numerous structures in the sporophytes of land plants. The highly modular structure of branched stems bearing sporangia in one of the earliest land plant fossils *Aglaophyton* and *Horneophyton* in the Rhynie chert biota is one of the oldest examples of modular structures in the sexual organs of land plants. The modular organization of shoot branches, leaves, and leaf-like structures in vascular plants is an example of this. Telomes in the sporophytes of early vascular plants like Devonian *Rhynia*, *Aglaophyton*, and *Cooksonia*, as well as the microphylls in species like *Rhynia*, *Zosterophylls*, *Psilophyton*, *Asteroxylon*, and *Lycopodium*, all reflect the modularity of plant architecture (Figure 3). The evolution of megaphylls in the Euphyllophyte clade, as theorized by Zimmermann [42,43], further illustrates the modularization process in plant evolution. The modular pattern in the sporophytes of all vascular plants is rooted in branching of the shoot and multiplication of sex organs in the common ancestor of the *Polysporangiophyta* clade. From the simple organization of individual sporangia at the end of each telome branch in *Cooksonia* and *Aglaophyton* to the complex structures like strobili in *Lycopodium* and *Equisetum*, and the diverse arrangements of sori and coenosori on fern fronds, modularization is evident in all sexual organs of extant vascular plants. This theme culminates in the cones of gymnosperms and the flowers of angiosperms, where modularization likewise plays a crucial role in reproductive success.

### 4.2. Subfunctionalization in Land Plants

Subfunctionalization is a process that utilizes the raw material of modularization to generate a variety of different modules with diverse functions—in other words, it constitutes the adoption of divergent morphological and functional attributes by existing modules, and the realization of division of labor at a level of organization. This definition is broad enough to incorporate the evolution of cell type diversity in the evolution of multicellularity, but its true value is revealed in its application to the evolution of diverse units above the cellular level—e.g., tissues and organs. This latter domain is virtually untouched by the standard ETI approach, since it characterizes the later stages of the transition to multicellularity. Paradigmatic examples from the animal kingdom include the modification of limbs (with its extreme form in the evolution of arthropods where limbs can carry out a truly broad range of functions) and more generally body segments, the emergence of distinct parts in the digestive tracts of several animal phyla with specific digestive and absorptive functions, and the evolution of different types of integumentary appendages such as sensory hairs and fur.

Subfunctionalization of the modular structure plays an important role in organization of the complexity and division of labor in both gametophyte and sporophytes of land plants. In early-diverging lineages of land plants, such as mosses, differentiation of algae-like cells of filamentous protonema (known as chloronemata) during early stages of gametophyte development into different cell types—such as caulonemata, which eventually differentiate into gametophore cells that form rhizoids and develop 3D body structures, including sexual organs—Refs. [44,45] exemplifies subfunctionalization in the gametophyte and results in the emergence of different functional parts throughout development. These resemble leaves, stems, and roots of the sporophyte of vascular plants, demonstrating a distinct division of labor. At the anatomical level, the clear subdivision of undifferentiated parenchymal cells produced by apical meristems into the epidermis, cortex, and central strand is evident in both the gametophyte and sporophyte stages of mosses. Within the central strand, specialized conducting cells known as hydroids that conduct water, and leptoids that conduct sugar, are distinctly present. In the gametophytes of mosses, this subfunctionalization mirrors the level seen in the sporophytes of vascular plants. Vascular plants are characterized by three tissue systems (dermal, ground, and vascular) that originate from three different meristematic cells (protoderm, ground meristem, and procambium). These three primary meristems are present in both root and shoot systems, produced by the Shoot Apical Meristem (SAM) in shoots and the Root Apical Meristem (RAM) in roots. This hierarchical complexity arose from an initial subfunctionalization of apical meristems in bryophytes, which likely evolved in response to spatial competition for light. This evolution led to the differentiation into SAM and RAM in vascular plants, essentially a division of labor between apical meristems for more efficient water and mineral acquisition through penetration into soil via the RAM, and enhanced light capture and spore dispersal via the arial growth of the SAM. This subfunctionalization in apical meristems enabled more efficient resource utilization and adaptation to diverse terrestrial environmental conditions.

A key aspect of the evolution of complex multicellularity in land plants is the shift in complexity and dominance between the sporophyte and gametophyte stages. In non-vascular plants, the highest level of complexity is observed in the haploid stage (gametophyte) of the cycle, more so than in any other land plants. Here, the gametophyte is the dominant and independent phase, with the sporophyte emerging as a dependent organ on the gametophyte. This heightened complexity of the bryophytes’ gametophyte represents a continuation of the increasing complexity in the haploid phase following its transition to multicellularity in the common ancestor of green plants. However, with the emergence of the multicellular diploid phase in land plants, the gametophyte began to simplify and become dependent, while the sporophyte, via co-option and re-deployment of an ancient algal homeodomain gene in the gametophyte [32,46], started to become independent and the dominant phase of the cycle. This evolutionary pattern is particularly pronounced in flowering plants, where the sporophyte attains greater complexity, and the gametophyte is reduced to microscopic, sporophyte-dependent, few-celled structures like the pollen grain (male gametophyte) and the ovule (female gametophyte). Therefore, it is essential to recognize this contrasting evolutionary trajectory in the complexity between the gametophyte and sporophyte phases in land plants.

In the evolution of multicellular complexity in land plants, a significant role is played by the subfunctionalization of structures within the sporophytes. Vascular plants, for instance, exhibit a modular organization in their shoot branches, leaves, and leaf-like formations, which have undergone multiple rounds of subfunctionalization. Early vascular plants such as Devonian *Rhynia*, *Aglaophyton*, and *Cooksonia* displayed foundational structures known as telomes (stems). As these plants evolved, telomes underwent subfunctionalization, leading to the development of specialized structures like microphylls, as seen in *Rhynia* (Rhyniophytina), *Zosterophylls* (Zosterophyllophyta), *Psilophyton* (Trimerophyta), *Asteroxylon*, and *Lycopodium* (Lycopodiophyta).

Zimmermann’s telome theory [42] offers a comprehensive framework for understanding this evolutionary trajectory, suggesting that structures like microphylls and megaphylls (true leaves) in vascular plants evolved through stages that are, we suggest, consistent with the processes of modularization, subfunctionalization, and integration. The theory posits that microphylls originated from the reduction of a single telome branch, while megaphylls evolved through the modification and integration of telomes. This theory, along with others like the enation theory [39] for microphyll evolution, highlights the diverse pathways through which these structures have evolved. The enation theory, for instance, suggests that microphylls emerged as new structures through the vascularization of tissue emergences on shoot flanks. Meanwhile, alternative theories like the reduction and sterilization theories propose that microphylls evolved from existing structures, either through reduction of lateral branch systems or successive sterilization of sporangia [19,47].

Despite these theories, there remains a lack of unequivocal support from paleobotanical, morphological, and phylogenetic studies for any single theory of microphyll evolution. However, comparative developmental genetic data, such as the conservation of KNOX/ARP expression patterns between lycophytes and seed plants, provide insights into leaf evolution and development and supports Zimmermann’s telome theory that leaves are in fact evolved from lateral branches [48,49]. This molecular evidence supports the telome theory, suggesting a homology of microphylls and megaphylls at the branch system level and indicating that the KNOX/ARP gene module was co-opted from a shoot-specific developmental program for leaf evolution in both lycophytes and seed plants, which mainly involved downregulation of KNOX and upregulation of APR [48,50]. However, LITTLE ZIPPER (ZPR) genes emerged from the duplication of a C3HDZ transcription factor (TF) (a crucial TF in determining form and growth of plant) in the common ancestor of euphyllophytes and created a unique regulatory module within the C3HDZ network specific to the euphyllophyte lineage and coincides with a period marked by rapid increase in the morphological complexity of euphyllophytes and may, therefore, be critical in the emergence of megaphyllous leaves [51]. Nonetheless, molecular data on the specific processes of megaphyll evolution proposed by the telome theory—overtopping, planation, and webbing—remain limited, as the evidence for the telome reduction process in microphyll evolution [50].

The evolution of leaves plays a pivotal role in land plants, as they are crucial in organizing the complexity within the sporophyte body plan, epitomizing the main processes of the TSC. Leaves exemplify modularity in plant architecture and demonstrate remarkable subfunctionalization, such as in the case of sporophylls in early euphyllophytes, the spines of cacti, and various flower parts including bracts, sepals, petals, stamens, and ovules, all of which are modified leaves resulting from subfunctionalization. Leaves also serve as a prime example of the integration step in TSC, where subfunctionalized leaves bearing sporangia integrate to form organs like strobili in *Lycopodium*, cones in gymnosperms, and flowers and fruits in angiosperms. The fact that almost all shoot appendages including floral organs are modified leaves, first suggested more than two centuries ago [52], was demonstrated experimentally two decades ago [53].

In early vascular plants, leaves underwent significant subfunctionalization, differentiating into fertile and sterile fronds. Initially, these leaves may have performed generalized functions, but over time, some evolved into fertile fronds specialized for reproduction and were adorned with sporangia for spore formation and dissemination. Others became sterile fronds, focusing solely on photosynthesis and other metabolic functions. This specialization not only enhanced reproductive efficiency but also maintained essential metabolic processes.

During the evolutionary history of vascular plants, heterospory, as another layer of subfunctionalization in the sexual life cycle of land plants, emerged multiple times within lycophytes, aquatic ferns, and progymnosperms [54,55]. Heterospory is a new layer of differences in the sexual life cycle that already involved male and female gametes in the cycle [56]. Fertile fronds further differentiated into microsporophylls and megasporophylls. Microsporophylls bear microsporangium and produce microspores, precursors to the male gametophyte, while megasporophylls contain megasporangium and develop megaspores, leading to female gametophytes. This specialization, which provides greater fitness per resource unit and decreases inbreeding [56], laid the foundation for the complex reproductive strategies seen in modern seed plants [57,58]. In gymnosperms, the distinction between microsporophylls and megasporophylls became more pronounced, with microsporophylls evolving into male cones bearing pollen and megasporophylls evolving into female cones housing ovules. In angiosperms, this evolutionary path reached a new level of complexity, with microsporophylls transforming into stamens and megasporophylls into carpels, comprising the ovary, style, and stigma. The ovary, a derivative of the megasporophyll, develops post-fertilization into fruit, protecting and aiding seed dissemination.

From the simple leaves of early vascular plants to the intricate floral arrangements of angiosperms, the process of subfunctionalization has underpinned the evolution of structural complexity in land plants by creating new functional units from modular structures, paving new routes for horizontal complexity in response to new terrestrial habitats.

### 4.3. Integration in Land Plants

Integration is the process whereby formerly separate units come together to compose a new cohesive unit. That this definition is strongly reminiscent of one of the ways in which ETIs—and even METs more generally—are defined is no coincidence. ETIs have been defined or at least characterized as events where formerly independently reproducing entities come together to form a new, higher-level unit that reproduces as a unit, thereby foregoing their own reproductive autonomy. Integration, in this sense, is a broader notion that subsumes this definition because it is agnostic about whether the constituent units give up their reproductive autonomy and only assumes that they form a higher-level structural (and typically also functional) unit. This allows the notion of integration to capture both the processes active in the earliest stages in a transition—e.g., where unicellular organisms begin to form a multicellular one—as well as those in later stages where groups of cells within an organism form new tissues, or groups of tissues form new organs. Integration builds on the outcomes of modularization and subfunctionalization as the undifferentiated or differentiated existing modules, often different kinds thereof, provide the raw materials for integration.

One of the most iconic integrative features in land plants is the flower. The flower exemplifies integration as multiple modules—sepals, petals, stamens, and carpels—come together to form a novel reproductive organ. These units, each with a unique role, integrate into a functional whole that not only facilitates pollination but also ensures the protection of the developing seeds. The seed itself, another evolutionary innovation, is another strong example of integration which packages the embryo, nutritive tissue, and a protective coat into a singular unit. This not only ensures the survival of the embryo in adverse conditions but also aids in its dispersal, thus expanding the plant’s geographical reach. Furthermore, vascular systems, consisting of the xylem and phloem, represent a significant case of integration of differentiated cells into conductive tissues. These transport systems allow plants to grow taller, accessing more sunlight and expanding their terrestrial reach. The emergence of the vascular cambium and cork cambium is a case of integration at work in this context, enabling secondary growth and adding girth to the plants.

## 5. Genomic Toolkits of Multicellularity in the TSC Context

The repetitive pattern of transitions in the complexity of multicellular lineages—i.e., repeated rounds of modularization, subfunctionalization, and integration—raises the question of whether these motifs in a series of morphological transformations are a result of fundamental underlying constraints, such as physical constraints, and/or common underlying genetic toolkits with similar functionality [27]. The relationship between genome size, gene number, and morphological complexity in multicellular organisms is intricate and not solely determined by the quantity of genetic material [59]. Despite sharing the same genome, the diverse cell types within multicellular organisms exhibit distinct structures and functions; furthermore, at least in the case of animals, much of the basic developmental genetic ‘toolkit’ of multicellularity has been found to be present in their closest unicellular group, highlighting the importance of transcriptional regulation in shaping complexity [46,60,61,62]. These evolutionary innovations have provided a foundation for the development of diverse cell types and organs with unique characteristics. Enhancers or cis-regulatory modules, which are binding sites for transcription factors [63], play pivotal roles in the regulatory networks of multicellular organisms, orchestrating spatio-temporal patterns of gene expression essential for development and morphological complexity [64]. Enhancers, scattered across the genome, integrate signaling inputs to specify gene expression patterns, allowing for combinatorial complexity in transcriptional regulation.

Whole-genome and small-scale duplications are recognized as crucial contributors to the evolution of functional innovation and morphological complexity and are the predominant means of generating new gene functions [65,66,67,68,69,70,71]. Following duplication, duplicated genes can experience a variety of evolutionary outcomes [67,71]. Typically, these genes become pseudogenes or inactive due to the accumulation of deleterious mutations, a process known as non-functionalization, occurring within a few million years [67,68,69,71]. On the other hand, duplicated genes theoretically can serve as a substrate for evolution due to their freedom from selective constraints, which allows for potential subfunctionalization and neofunctionalization [67,71,72]. Following the whole-genome duplication, that we can call modularization at the genomic level since it increases the number of functional units in the genome, the retention rates of duplicated genes or modules differ. For instance, genes involved in transcriptional regulation, signal transduction, and development are retained at higher rates compared to other functional categories [69,73,74]. Conversely, after large-scale duplication and the emergence of polyploid organisms, many duplicates become non-functional or are deleted over time, leading to a reduction in genome size and extensive genome reorganization. This process, known as diploidization, converts polyploids back into diploids over several million years. Species that undergo diploidization following polyploidization are termed palaeopolyploids. Indeed, all existing angiosperms are palaeopolyploid [72], and it is estimated that 15% of angiosperm and 31% of fern speciation events are accompanied by an increase in ploidy, and most plant species, if not all, have a polyploid ancestor [75]. The whole-genome duplication, however, is generally less frequent in animals than in plants [76,77,78,79,80].

It must be pointed out here that the term subfunctionalization is used in the context of genomics to refer to the process where duplicate genes divide the original function between them, with each retaining a part of the ancestral function; and its allied term neofunctionalization refers to cases where one of the duplicate genes acquires a new function that was not present in the ancestor, while the other retains the original function. Nevertheless, the flexibility of the notion of subfunctionalization in the context of TSCs enables it to subsume both these genomic notions, alongside many other processes discussed so far, under its general rubric. Furthermore, the process of duplication itself falls under the notion of modularization in its broad sense in the TSC context. Together, these highlight the applicability of the TSC approach to the genomic level, as previously discussed by Nejad Kourki [9].

The evolutionary history of multicellularity in land plants is filled with instances where gene duplication, particularly of transcription factors, followed by subfunctionalization or neofunctionalization of paralogous genes, has driven innovations in morphological complexity. For instance, the origin of flowers involves various MADS box transcription factor complexes, with their evolution attributed to ancient and recent gene duplications followed by subfunctionalization of paralogous genes. In orchids, the most species-rich and florally diverse angiosperm family, subfunctionalization of duplicated B genes has led to the development of three petal-like organs: three outer tepals, two inner tepals, and a highly modified lip [81].

Another example is modularization in leaves through the dissection of simple leaves into leaflets in Brassicaceae, which has been shown to be a result of gene duplication of *LMI1* (*LATE MERISTEM IDENTITY 1*)-type sequences, which created the *REDUCED COMPLEXITY* (*RCO*) homeodomain protein, which is required for leaflet development [82], and its evolutionary changes explain leaf margin dissection patterns and temperature-dependent leaf margin phenotypic plasticity in Brassicaceae [83].

Class III Homeodomain-Leucine Zipper (Class III HD-Zip) Genes are another important group of transcription factors in land plants, which displays functions in sporophyte apical meristem formation, vascular patterning, and the adaxial polarity of leaves [84]. Multiple duplication events throughout the evolution of land plants increased the number of genes in these and other developmentally important families of transcription factors, such as the APETALA2 Family, Class I KNOX gene, and TCP Family, which all are associated with increased morphological complexity through subfunctionalization and neofunctionalization [50], corresponding to subfunctionalization in the context of TSCs, with modularization operating in the background.

Another intriguing example is the evolution of stomata and their closure mechanisms, which originated once in the common ancestor of land plants [85]. However, the gene families involved in stomatal closure underwent multiple duplications, mainly in the ancestor of seed plants, which preceded the neo- or subfunctionalization of these genes [86]. For example, genes involved in signaling through certain potassium and anion channels existed before land plants emerged and expanded in the ancestor of seed plants [86].

## 6. Conclusions

We live in an era marked by an ever-growing abundance of scientific findings spanning an unprecedented proliferation of methods and approaches in a vast landscape of research programs. This means that, more than ever before, we are in need of theoretical approaches to organize these findings into cohesive bodies of knowledge that not only bring about understanding but also play the more practical role of providing cross-linkages between scientific fields, facilitating theoretical, conceptual, and methodological transfer between different research programs and help generate new testable hypotheses across the board (for a recent discussion along these lines, see [87]). To do this, we need unifying conceptual frameworks that span the boundaries of existing methodologies and areas of study by utilizing abstract concepts that work together cohesively. The TSC approach provides one such framework, and the processes of modularization, subfunctionalization, and integration operate as abstract and flexible organizing concepts that can unify such diverse evolutionary events as the emergence of a new gene that can then go on to perform a new function to the emergence of flowers from existing units, and even the emergence of multicellularity itself.

It must be pointed out, though, that the aim here has not been to fully catalog key examples of the processes of TSC in land plants, nor to map them accurately or comprehensively onto the phylogeny of plants. And, of course, we have not—and indeed could not have—covered every aspect of the evolution of complex multicellularity in plants and how it is distinct from such cases elsewhere (e.g., animals); for example, we have not touched on the importance of intercellular communication via plasmodesmata vs. tight and gap junctions in animals (see [88,89,90]). Rather, our aim has been to highlight the viability and potential explanatory value of the TSC approach in the context of land plant evolution in a manner analogous to the attempt by Nejad Kourki [9] in the context of animal evolution. This highlights the organizing role that TSC is intended to play in explaining the evolution of structural complexity, which is to pick out key events that are at once unique and similar to others across taxa as well as levels of organization. Our hope is that this line of work is taken further in the future not only in the context of animal and plant evolution but also in other contexts that demonstrably involve the evolution of structural complexity, such as the classic ETIs including the origin of life on Earth and the emergence of eukaryotes, as well as the evolution of eusocial superorganisms and, perhaps, human societies.

## Figures and Tables

**Figure 1 genes-15-01472-f001:**
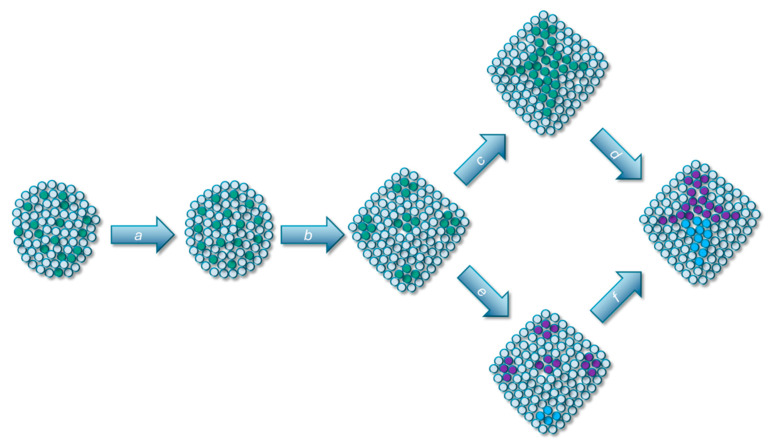
Abstract representation of the three processes of the evolution of structural complexity. (a) Modularization resulting in spatial regularity of the specialized cells (purple) in the form of rosettes in a colonial-grade organism. (b) Localized integration resulting in formation of pockets of specialized cells, with potential synergistic functionality. (c,f) Integration. (d,e) Subfunctionalization resulting in two novel cell types (red and blue), each spatially localized to perform their divergent functions. Modified from [9].

**Figure 2 genes-15-01472-f002:**
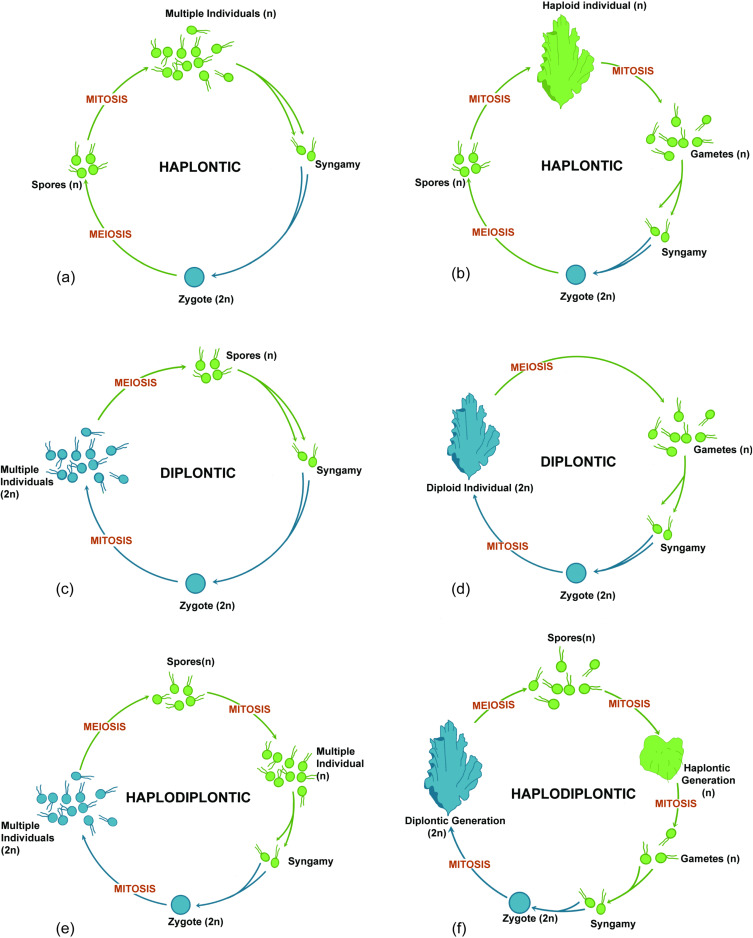
Schematic comparison between (**a**,**b**) haplontic, (**c**,**d**) diplontic, and (**e**,**f**) haplodiplontic life cycles in unicellular (**a**,**c**,**e**) and multicellular (**b**,**d**,**f**) organisms (illustrated by Shirin Sabeti).

**Figure 3 genes-15-01472-f003:**
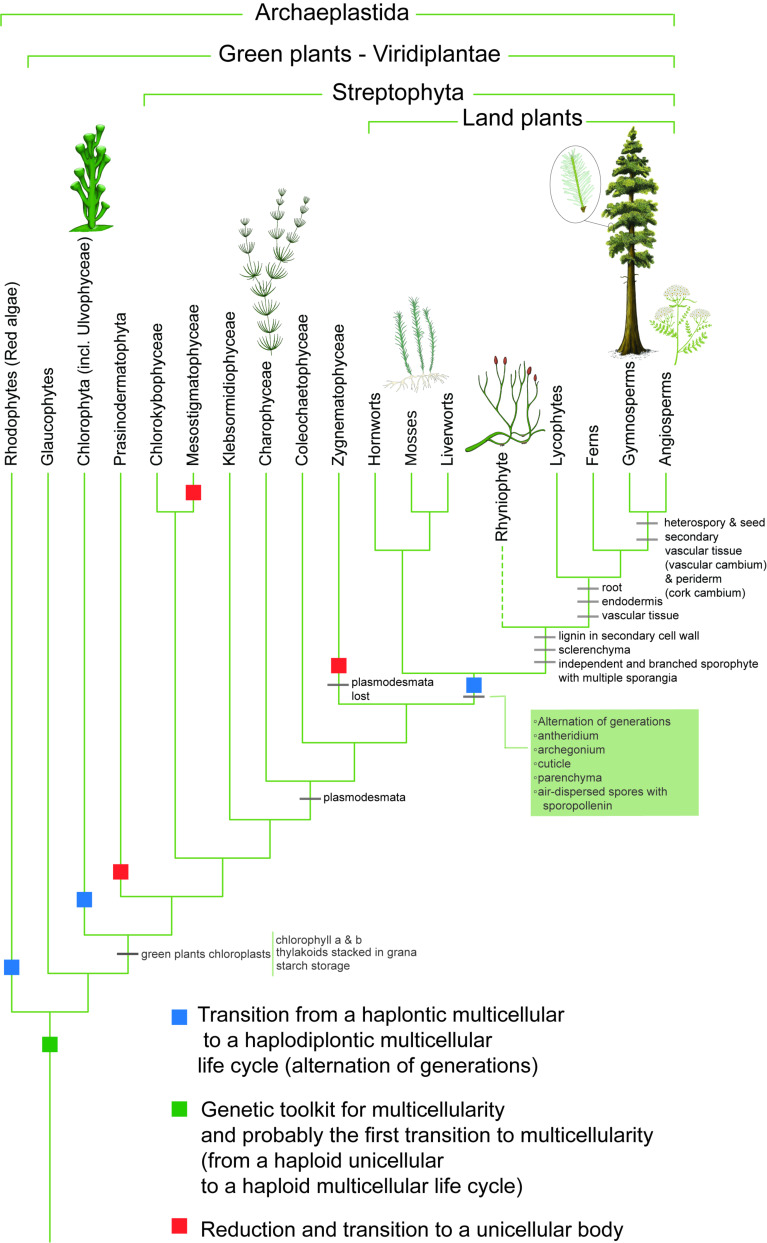
Transition to multicellularity in *Archaeplastida*: at least four independent transitions to a haplodiplontic life cycle occurred in Archaeplastida—two in red algae, one in *Ulvophyceae*, and one in land plants (phylogeny modified from [40]; illustrations by Shirin Sabeti).

**Table 1 genes-15-01472-t001:** This table summarizes key differences between the ETI and TSC approaches. Whereas the principal explanatory aim of the ETI approach is the evolution of new units of selection, especially earlier stages thereof (formation and transformation; see [8]), the principal aim of the TSC approach is the explaining the evolution of new units of organization more generally, which includes units of selection more specifically, and with a special focus on later stages (transformation [8]). And the two approaches also differ in their explanatory tools: while the ETI approach primarily utilizes population genetics and game theoretic models, the TSC approach utilizes the three processes discussed in this paper (as well as [9]) against the backdrop of the transfer of functions to the higher level of organization.

Theoretical Approach	Explanatory Aim	Example Target Phenomena	Core Explanatory Tools
**Evolutionary Transitions in Individuality (ETI)**	Evolution of new units of selection (formation and maintenance)	Protocell Eukaryotic cell Multicellularity Eusociality	Population genetics Game theoretic models
**Transitions in Structural Complexity (TSC)**	Evolution of new units of organization (transformation)	All of the above, and: Organelles Tissues Organs Etc.	• Three processes: ○ Modularization ○ Subfunctionalization ○ Integration • Transfer of function

## Data Availability

The data presented in this study are available in this article.

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
