# Peer review of "The Evolution of Complex Multicellularity in Land Plants"

_genes, 2024, doi:10.3390/genes15111472_

Round 1
Reviewer 1 Report
Comments and Suggestions for Authors
This manuscript has an ambitious goal to describe a framework for the evolution of complex multicellularity in land plants. Overall, I think the subject matter should be of broad interest to readers of botany (and general biology). While the text is generally well-written, free of grammar and spelling issues, it's sometimes difficult to follow due to technical abstract jargons. I think more figures would help bring the ideas to readers better. The following are a few specific comments and suggestions.
1. It would be great to have a figure illustrating the ETI framework and TSC approach, highlighting their major components and novelty.
2. In the current figure 1, many words have letters that are squeezed together and it's difficult to read them. Please fix.
3. It isn't not clear to me the number of plant transitions from aquatic to terrestrial environments. Was it only once? Did all the transitions involve multicellular aquatic plants to multicellular terrestrial plants? Was there any independent origin of multicellular land plants from unicellular land organisms?
4. Was there any transitions back from terrestrial to aquatic environments? How changes in their structures and genomes may tell us about the origins of multicellularity in land plants.
4. A table describing the (likely) main characteristics of the early lineages of multicellular aquatic plants and multicellular terrestrial plants would be very helpful in orienting readers while going this manuscript. This should include the novel traits land plants have evolved.
5. A major deficiency in the manuscript is the limited inclusion and synthesis of recent comparative genomics across the plant kingdom. The current description is very rudimentary. The authors are strongly encouraged to add a section on this topic, highlighting the candidate genes involved in the evolution of various organs unique to land plants.
line 76: can the senior author of this manuscript have a personal communication within him/herself?
Comments on the Quality of English Language
The English is fine
Author Response
Comment 1: It would be great to have a figure illustrating the ETI framework and TSC approach, highlighting their major components and novelty.
Response1: We added a figure and a table illustrating this.
Comment 2: In the current figure 1, many words have letters that are squeezed together and it's difficult to read them. Please fix.
Response1: Fixed.
Comment 3: It isn't not clear to me the number of plant transitions from aquatic to terrestrial environments. Was it only once? Did all the transitions involve multicellular aquatic plants to multicellular terrestrial plants? Was there any independent origin of multicellular land plants from unicellular land organisms?
Response 3: Transition from aquatic to terrestrial environment was not a center of focus in this paper, that is why we did not discuss the number of transitions. However, if we are talking only about embryophyta, we should say it only occurred once. Terrestrial land plants are the result of a single transition to land and from probably a multicellular aquatic ancestor.
Comment 4: Was there any transitions back from terrestrial to aquatic environments? How changes in their structures and genomes may tell us about the origins of multicellularity in land plants.
Response 4:Yes, there are multiple instances of transitioning back to aquatic environments in land plants, like aquatic ferns and aquatic flowering plants such as water lily, azolla, ceratophyllum, etc. In all cases we generally can see major reduction in morphological and genomic complexity along with gaining some morphological and physiological novelties for adaptation to aquatic environments.
Comment 5:. A table describing the (likely) main characteristics of the early lineages of multicellular aquatic plants and multicellular terrestrial plants would be very helpful in orienting readers while going this manuscript. This should include the novel traits land plants have evolved.
Response 5: We added the synapomorphies for land plants to the phylogenetic tree in the fig.2
Comment 6: A major deficiency in the manuscript is the limited inclusion and synthesis of recent comparative genomics across the plant kingdom. The current description is very rudimentary. The authors are strongly encouraged to add a section on this topic, highlighting the candidate genes involved in the evolution of various organs unique to land plants.
Response 6: We have provided a section for the genomic toolkit involved in multicellularity. However, as it is highlighted in the title of this section now, we tried to assess the evolution of the genetic toolkit in the context of TSC and provide some examples following TSC patterns and leading to structural complexity.
Comment 7: line 76: can the senior author of this manuscript have a personal communication within him/herself?
Response 7: Removed.
Reviewer 2 Report
Comments and Suggestions for Authors
This paper is a highly theoretical exploration into the origins of multicellularity in land plants. Overall I found the work interesting but rather wordy and repetitious with a lot of fundamental basic information missing. The text contains a lot of teleology which needs to be revised eg line 339 conducting cells known as hydroids, for water conduction, and leptoids, for sugar conduction.
The greatest omission is the failure of the authors to consider the most important feature of multicellularity namely intercellular communication: plasmodesmata in plants versus tight and gap junctions in animals plus their absence in callus and carcinomas. This has to be considered in an article like this.
It requires a much wider literature review. Perhaps the authors should consult someone more aware of this than they appear to be.
Specifics;-
Line 280 ‘antheridial and archegonial heads stand out’. Very poor wording for the elevated carpocephala found in may complex thalloid liverworts.
Line 287 this needs references adding-see papers by L Dolan et al
Line 289 the primary role of rhizoids is nutrient uptake
Line 328 Protonemata are not the same as filamentous algae as they elongate by tip not intercalary growth
Line 333 Bryophytes lack roots
Comments on the Quality of English Language
Revise to get rid of teleology. Otherwise OK
Author Response
Comment 1: This paper is a highly theoretical exploration into the origins of multicellularity in land plants. Overall I found the work interesting but rather wordy and repetitious with a lot of fundamental basic information missing. The text contains a lot of teleology which needs to be revised eg line 339 conducting cells known as hydroids, for water conduction, and leptoids, for sugar conduction.
Response1: Thanks for the appreciation! Regarding teleology: we do not intend the use of teleological terminology to suggest or imply a strong teleological stance, and only utilise it as a shorthand. Doing so is perfectly in line with common practice in biological writing, where the teleological language surrounding functions of traits (e.g. the function of hydroids is water conduction) is merely a shorthand for talking about what that trait has been adapted for under natural selection. Since this is common practice, we believe that specifying this shorthand nature of teleological language is unnecessary and detracts from conciseness.
Comment 2: The greatest omission is the failure of the authors to consider the most important feature of multicellularity namely intercellular communication: plasmodesmata in plants versus tight and gap junctions in animals plus their absence in callus and carcinomas. This has to be considered in an article like this.
It requires a much wider literature review. Perhaps the authors should consult someone more aware of this than they appear to be.
Response1: We fully recognise the importance of taking into account intercellular communication in relation to the evolution of complex multicellularity, and we also acknowledge that this is something outside of our expertise. Nevertheless, no analysis of any topic can ever cover its every aspect, and we would be more than happy to see the growth of the discussion carried out by an expert on intercellular communication in this context. We have therefore now explicitly specified this in the text (in the conclusion).
Comment 3: Line 280 ‘antheridial and archegonial heads stand out’. Very poor wording for the elevated carpocephala found in may complex thalloid liverworts environments.
Response 3: Right! Thank you for pointing out this issue. We improved the wording of this part.
Comment 4: Line 287 this needs references adding-see papers by L Dolan et al
Response 4: We added a reference from Dolan to this part.Comment 5: Line 289 the primary role of rhizoids is nutrient uptake
Response 5: Thank you for your valuable comment regarding the primary role of rhizoids. We acknowledge the traditional view that rhizoids primarily function in nutrient uptake, as you noted. However, we believe it is important to emphasize that rhizoids serve multiple roles, particularly anchorage, as well as nutrient and water uptake, depending on the species.
Our description aligns with the literature, which asserts that anchorage is often considered the primary role of rhizoids, especially in bryophytes and non-seed vascular plants. According to Duckett et al. (1998), Goffinet et al. (2008), and Crandall-Stotler et al. (2009), rhizoids primarily contribute to attachment to the substrate. Additionally, studies on liverworts, mosses, and filmy ferns show that rhizoids form adhesive structures for attachment (Duckett et al., 1991, 1996; Haberlandt, 1914). While rhizoids also facilitate water and nutrient absorption, these functions vary across species and are not always their primary role, as suggested by Proctor (1984, 2000) and other studies.
In conclusion, while nutrient uptake is a significant function of rhizoids in some species, anchorage remains their primary role in many bryophytes, as supported by the literature. Thus, our description aims to reflect this dual functionality. We hope this clarifies our reasoning behind the statement.
Comment 6: Line 328 Protonemata are not the same as filamentous algae as they elongate by tip not intercalary growth
Response 6: Yes, protonemata are different from filamentous algae. We only highlighted the structural similarity between them, not that they are the same! Additionally, apical growth is not exclusive to land plants, as tip growth can also be observed in various streptophyte algae. These algae also share other similarities with land plants, such as 2D and 3D growth in their protonema-like structures (see Buschmann, H., 2020). We changed the wording, added some details along with two new references.
Comment 7: Line 333 Bryophytes lack roots
Response 7: Yes, bryophytes lack roots but their gametophytes have rhizoids that resemble roots in the sporophytes in vascular plants. And we tried to acknowledge the similarity in the subfunctionization of gametophyte and sporophyte.
Comment 8 (Comments on the Quality of English Language): Revise to get rid of teleology. Otherwise OK
Response 8:
Round 2
Reviewer 2 Report
Comments and Suggestions for Authors
Please see attached file

Author Response
Considering the major revision recommended in my original review of this paper I am saddened that the authors have taken these so lightly and produced a barely changed paper.
The text is far too wordy and could be reduced by at least 30% without any loss of vital information. As the paper is theoretical it is vital that it is built on a firm knowledge base. This is lacking.
PREVIOUS REVIEW
Comment 1: This paper is a highly theoretical exploration into the origins of multicellularity in land plants. Overall I found the work interesting but rather wordy and repetitious with a lot of fundamental basic information missing. The text contains a lot of teleology which needs to be revised eg line 339 conducting cells known as hydroids, for water conduction, and leptoids, for sugar conduction.
Response1: Thanks for the appreciation! Regarding teleology: we do not intend the use of teleological terminology to suggest or imply a strong teleological stance, and only utilise it as a shorthand. Doing so is perfectly in line with common practice in biological writing, where the teleological language surrounding functions of traits (e.g. the function of hydroids is water conduction) is merely a shorthand for talking about what that trait has been adapted for under natural selection. Since this is common practice, we believe that specifying this shorthand nature of teleological language is unnecessary and detracts from conciseness.
Reviewer’s response. Teleology is totally unacceptable in scientific writing and can easily be removed without increasing the word count.eg Line 365 hydroids for water conduction : change to hydroids that conduct water and leptoids that conduct sugars. Believers in intelligent design would love this teleology!
Response 1: though as mentioned before mere teleological language does not actually iply teleology and is widely used (despite the reviewer’s insistence), we have modified our wording so as to avoid any possible confusion.
Comment 2: The greatest omission is the failure of the authors to consider the most important feature of multicellularity namely intercellular communication: plasmodesmata in plants versus tight and gap junctions in animals plus their absence in callus and carcinomas. This has to be considered in an article like this. It requires a much wider literature review. Perhaps the authors should consult someone more aware of this than they appear to be.
Response 2: We fully recognise the importance of taking into account intercellular communication in relation to the evolution of complex multicellularity, and we also acknowledge that this is something outside of our expertise. Nevertheless, no analysis of any topic can ever cover its every aspect, and we would be more than happy to see the growth of the discussion carried out by an expert on intercellular communication in this context. We have therefore now explicitly specified this in the text (in the conclusion).
Reviewer’s response Without a proper consideration of intercellular communication the whole article becomes somewhat meaningless. Plasmodesmata are included in Fig 3 but not discussed in the text . An article like this must include a discussion of symplastic and apoplastic transport as a fundamental difference between plants and animals. The revised text is lazy in the extreme: either do the necessary research or enlist someone who knows this subject. Fig 1 omits channels of communication which are fundamental to the distinction between colonies and multicellular organisms. The present writing is not up to the standard expected for the journal.
Response 2: We appreciate the reviewer’s engagement and agree that intercellular communication, including symplastic and apoplastic transport, is a significant aspect of multicellularity. However, as our study’s focus lies elsewhere, a full exploration of these processes is beyond our intended scope. Our aim was to concentrate on broader evolutionary patterns and transitions rather than delve into detailed cellular mechanisms, which could quickly broaden the scope into areas where expertise is essential. We believe that focusing on such patterns provides valuable insights while also allowing others with specialized knowledge to further enrich the conversation.
The inclusion of plasmodesmata in Fig. 3 was meant to acknowledge intercellular communication's relevance in multicellularity without compromising the article's scope. We have also now explicitly referenced this limitation in the text to clarify our intent and maintain transparency with readers. We hope this approach provides the appropriate balance, considering our focus on evolutionary principles rather than an exhaustive treatment of cellular communication channels.
Comment 3: Line 280 ‘antheridial and archegonial heads stand out’. Very poor wording for the elevated carpocephala found in may complex thalloid liverworts environments.
Response 3: Right! Thank you for pointing out this issue. We improved the wording of this part. Reviewer’s response Fine
Comment 4: Line 287 this needs references adding-see papers by L Dolan et al
Response 4: We added a reference from Dolan to this part.
Reviewer’s response Fine
Comment 5: Line 289 the primary role of rhizoids is nutrient uptake
Response 5: Thank you for your valuable comment regarding the primary role of rhizoids. We acknowledge the traditional view that rhizoids primarily function in nutrient uptake, as you noted. However, we believe it is important to emphasize that rhizoids serve multiple roles, particularly anchorage, as well as nutrient and water uptake, depending on the species. Our description aligns with the literature, which asserts that anchorage is often considered the primary role of rhizoids, especially in bryophytes and non-seed vascular plants. According to Duckett et al. (1998), Goffinet et al. (2008), and Crandall-Stotler et al. (2009), rhizoids primarily contribute to attachment to the substrate. Additionally, studies on liverworts, mosses, and filmy ferns show that rhizoids form adhesive structures for attachment (Duckett et al., 1991, 1996; Haberlandt, 1914). While rhizoids also facilitate water and nutrient absorption, these functions vary across species and are not always their primary role, as suggested by Proctor (1984, 2000) and other studies. In conclusion, while nutrient uptake is a significant function of rhizoids in some species, anchorage remains their primary role in many bryophytes, as supported by the literature.
Thus, our description aims to reflect this dual functionality. We hope this clarifies our reasoning behind the statement.
Reviewer’s response Better but could still be improved. The evidence indicates water and nutrient uptake as primary. Anchorage is almost certainly secondary
Comment 6: Line 328 Protonemata are not the same as filamentous algae as they elongate by tip not intercalary growth
Response 6: Yes, protonemata are different from filamentous algae. We only highlighted the structural similarity between them, not that they are the same! Additionally, apical growth is not exclusive to land plants, as tip growth can also be observed in various streptophyte algae. These algae also share other similarities with land plants, such as 2D and 3D growth in their protonema-like structures (see Buschmann, H., 2020). We changed the wording, added some details along with two new references.
Reviewer’s response OK And please note that angelized Latin takes lower case eg streptophyte
Comment 7: Line 333 Bryophytes lack roots
Response 7: Yes, bryophytes lack roots but their gametophytes have rhizoids that resemble roots in the sporophytes in vascular plants. And we tried to acknowledge the similarity in the subfunctionization of gametophyte and sporophyte.
Reviewer’s response OK
Reviewer’s response Rhizoids do not resemble roots any more than do bird, bat and insect wings. Please change and cite works by Raven, Edwards and Kenrick .
Response: While it is true, as the reviewer points out, that rhizoids and roots are not homologous structures (we acknowledge this, despite shared pathways underpinning the development of roots and rhizoids), there is no point at which we have claimed homology. A claim that two things resemble each other is a much weaker claim than a claim of homology. We are interested in convergent adaptive evolution as well as tracing homologous characters in this paper, and have been careful not to confuse a claim of resemblance (analogy) with a claim of sameness (homology).
Additional reviewer’s response
Line 622-625 on stomata. This is a gross oversimplification and needs better consideration of what is a highly contentious issue. Please consult the works of Brodribb and McAdam.Response: In these lines we have merely acknowledged the limitations of our study, and it is not quite fair to characterise this as a gross oversimplification. Nonetheless, we have now cited the papers suggested by the reviewer.
Round 3
Reviewer 2 Report
Comments and Suggestions for Authors
The authors have made just about satisfactory responses to my comments and the paper is now in much better shape.
The absence of consideration of intercellular coammunication Line 593 needs to be in the introduction NOT the discussion.
Considering the one of the author's principle research interests I am most surprised by the failure to consider intercellular communication . It is sad that this paper lacks this dimension.
Author Response
We have made all relevant changes according to the last set of comments.